# Comparative analysis of virulence determinants, phylogroups, and antibiotic susceptibility patterns of typical versus atypical Enteroaggregative *E. coli* in India

Vinay Modgil[1], Jaspreet Mahindroo[1], Chandradeo Narayan[1], Manmohit Kalia[1], Md Yousuf[1], Varun Shahi[1], Meenakshi Koundal[1], Pankaj Chaudhary[2], Ruby Jain[3], Kawaljeet Singh Sandha[3], Seema Tanwar[4], Pratima Gupta[5], Kamlesh Thakur[6], Digvijay Singh[7], Neha Gautam[7], Manish Kakkar[8], Bhavneet Bharti[2], Balvinder Mohan[1], Neelam Taneja[1] *

1 Department of Medical Microbiology, Postgraduate Institute of Medical Education and Research (PGIMER), Chandigarh, India, 2 Department of Pediatrics Medicine, Postgraduate Institute of Medical Education and Research, Chandigarh (PGIMER), India, 3 Civil Hospital Manimajra, Chandigarh, India, 4 Civil Hospital Ambala, Haryana, India, 5 Department of Microbiology, All India Institute of Medical Sciences (AIIMS), Rishikesh, Uttarakhand, India, 6 Department of Microbiology, Dr. Rajendra Prasad Government Medical College Kangra (RPGMC), Himachal Pradesh, India, 7 Department of Microbiology, Indira Gandhi Medical college (IGMC), Shimla, Himachal Pradesh, India, 8 Public Health Foundation of India, Delhi, India

* drneelampgi@yahoo.com

**Data Availability Statement:** All relevant data are within the manuscript and its Supporting Information files.

## Abstract

Enteroaggregative *Escherichia coli* (EAEC) is an evolving enteric pathogen that causes acute and chronic diarrhea in developed and industrialized nations in children. EAEC epidemiology and the importance of atypical EAEC (aEAEC) isolation in childhood diarrhea are not well documented in the Indian setting. A comparative analysis was undertaken to evaluate virulence, phylogeny, and antibiotic sensitivity among typical tEAEC versus aEAEC. A total of 171 EAEC isolates were extracted from a broad surveillance sample of diarrheal (N = 1210) and healthy children (N = 550) across North India. Polymerase chain reaction (PCR) for the *agg*R gene (master regulator gene) was conducted to differentiate tEAEC and aEAEC. For 21 virulence genes, we used multiplex PCR to classify possible virulence factors among these strains. Phylogenetic classes were identified by a multiplex PCR for *chu*A, *yja*A, and a cryptic DNA fragment, TspE4C2. Antibiotic susceptibility was conducted by the disc diffusion method as per CLSI guidelines. EAEC was associated with moderate to severe diarrhea in children. The prevalence of EAEC infection (11.4%) was higher than any other DEC group (p = 0.002). tEAEC occurrence in the diarrheal group was higher than in the control group (p = 0.0001). tEAEC strain harbored more virulence genes than aEAEC. *ast*A, *aap*, and *agg*R genes were most frequently found in the EAEC from the diarrheal population. Within tEAEC, this gene combination was present in more than 50% of strains. Also, 75.8% of EAEC strains were multidrug-resistant (MDR). Phylogroup D (43.9%) and B1 (39.4%) were most prevalent in the diarrheal and control group, respectively. Genetic analysis revealed EAEC variability; the comparison of tEAEC and aEAEC allowed us to better

**Funding:** This research was supported partially from PHFI grant for capacity research building within India Research Initiative on Peri-Urban Human-Animal Environment Interface, Indian Council of Medical Research (ICMR) grant to the PGIMER (Grant Number: 5/8-1(37) 2012-13 ECDII) and University Grant Commission (UGC) (Sr. No. 20614305077, Ref No: 22/06/2014(i)EU-V). The funders had no role in study design, data collection and analysis, decision to publish, or preparation of the manuscript.

**Competing interests:** The authors have declared that no competing interests exist.

understand the EAEC virulence repertoire. Further microbiological and epidemiological research is required to examine the pathogenicity of not only typical but also atypical EAEC.

## Author summary

Enteroaggregative *E. coli* (EAEC) are an increasingly important cause of diarrhea. *E. coli* belonging to this category cause watery diarrhea, which is often persistent and can be inflammatory. It is also associated with traveler's diarrhea in children and adults in middle and high-income countries. EAEC are defined by their ability to adhere to epithelial cells in a characteristic stacked brick-like pattern. However, the identification of these pathogenic strains remains elusive because of its heterogeneous nature. Genes that could contribute to the pathogenicity of EAEC encode adhesions, toxins, and other factors. Due to the heterogeneity of EAEC strains and differing host immune responses, not all EAEC infections are symptomatic. A critical factor in both recognizing EAEC pathogenesis and defining typical EAEC (tEAEC) strains is AggR, a transcriptional control for many EAEC virulence genes. The central role of *agg*R in virulence confers a strong priority to understand its pathogenicity. To identify EAEC, the CVD432 probe has been used. The CVD432 is a DNA probe from pAA plasmid of EAEC, has been reported to be specific for the detection of EAEC. The lack of sensitivity comes from the genetic heterogeneity of the EAEC strains and the wide geographic dispersal of strains. In our study, we performed a large surveillance of EAEC from North India among the pediatric population. Samples were collected by the microbiology staff at the Postgraduate Institute of Medical Education and Research (PGIMER) and referral system labs in Chandigarh (Manimajra), Punjab (Ludhiana), Haryana (Panchkula and Ambala Cantt), Himachal Pradesh (Hamirpur, Shimla, and Tanda), and Uttarakhand (Rishikesh, Rudrapur, and Haridwar)]. PGIMER is the largest tertiary care hospital in North India and serves patients from across Punjab, Jammu and Kashmir, Himachal Pradesh and Haryana. EAEC infections were detected using molecular methods. In our finding, *ast*A, *aap*, and *agg*R genes were most frequently found in the EAEC from the diarrheal population. Within tEAEC, this gene combination is present in more than 50% of strains and helps to differentiate tEAEC from aEAEC. Our collection of EAEC strains helps in finding an appropriate marker for the early detection of EAEC. Our signature sequence (*ast*A, *aap*, and *agg*R) will be ideal as focus genes for EAEC identification, as well as tEAEC and aEAEC. The multidrug resistance (MDR) was observed in 75.8% of the EAEC strains. tEAEC exhibits resistance to a greater number of antibiotics with respect to aEAEC. The phylogenetic analysis revealed that EAEC phylogeny is diverse and dispersed in all the phylogroups.

## Introduction

Diarrhea has been reported as the second most common cause of mortality all over the world in children less than 5 years of age, causing approximately 1.5 million pediatric deaths per year (WHO/UNICEF, 2009). Diarrheogenic *E. coli* (DEC) are the strains of *E. coli* that, by certain virulence factors, lead to diarrhea in humans. Currently, DEC has been categorized into six major pathotypes, including Enteropathogenic *E. coli* (EPEC), Enterotoxigenic *E. coli* (ETEC), Enteroaggregative *E. coli* (EAEC), Enteroinvasive *E. coli* (EIEC), Enterohemorrhagic *E.coli* (EHEC), and Diffusely Adherent *E. coli* (DAEC). EAEC is amongst the major bacterial causes of diarrhea in children worldwide. EAEC is thought to be the most recently identified

pathotype among DEC [1]. This pathotype is significantly associated with acute and persistent diarrhea, malnutrition in children and HIV infection, and traveler's diarrhea [2–5]. However, the distribution of EAEC varies from one geographic location to another [6].

EAEC pathogenesis study is limited due to its strain heterogeneity. The clinical spectrum of disease in children varies from a subclinical infection/intestinal colonization, which leads to acute and persistent diarrhea. The current pathogenesis model of the EAEC includes the following three stages. 1) adherence to the epithelial layer through adherence of fimbriae (AAF) or other adherence factors to the organism; 2) increased production of mucus by bacteria and intestinal cells that encrusts EAEC on the surface of the mucosal lining, and 3) an inflammatory response with cytokine release that may occur [7]. EAEC basic approach seems to include penetration of the intestinal mucosa, probably mainly that of the colon, accompanied by the secretion of enterotoxins and cytotoxins [8]. Human-intestinal research shows that EAEC causes moderate to severe mucosal damage [9]. EAEC strains usually increase mucus production, trapping the bacteria in a bacterium-mucus biofilm [10]. A thick biofilm may be linked to its ability to cause chronic colonization and diarrhea [10]. Some EAEC strains, while forming a mucous biofilm, cause cytotoxic effects on the intestinal mucosa. Clinical features of EAEC diarrhea include watery, mucoid, low-grade secretory diarrhea, sometimes grossly bloody stools, and little to no vomiting, and these are usually well-defined in sporadic cases, outbreaks, and controlled human infection volunteer models [11–14].

Most EAEC strains colonize intestinal mucosa with the help of adhesion fimbriae (AAF) [15]. These AAFs include at least four major antigenic variants (AAF/I-AAF/IV). These AAFs are governed by an AraC/XylS family activator named AggR [16]. AggR also aids in the gene expression encoding dispersin (*aap* gene), Aat (dispersin translocator), and the so-called chromosomal cluster (Aai) encoding a type VI secretion system in EAEC [17]. EAEC is classified as typical EAEC (tEAEC) and atypical EAEC (aEAEC) based on the presence or absence, respectively, of the *aggR*. In most EAEC epidemiological studies, tEAEC strains were mainly targeted, whereas aEAEC (AggR-regulon-negative EAEC) strains were excluded. While many investigators have found a strong association of tEAEC with diarrhea, the pathogenicity of aEAEC has not been clearly defined. Nevertheless, aEAEC has also been identified as a food-borne pathogen and has been significantly associated with outbreaks [18]. Factors not regulated by AggR include Air adhesin, EilA (HilA like regulator), EAEC heat-stable toxin EAST-1 (encoded by the *ast*A gene), and a group of toxins classified as serine protease autotransporters of Enterobacteriaceae (SPATE) [16].

Phylogenetically, SPATEs are classified into two groups [19]. Class 1 SPATE members are cytotoxic to the epithelium. These class 1 SPATEs are found in EAEC strains and include plasmid-encoded toxin (Pet) and its homologs, (Sat) autotransporter toxin, and (SigA) IgA protease-like homolog. Class 2, or non-cytotoxic SPATEs, contain Pic, a mucinase that facilitates EAEC intestinal colonization [20] and SepA, which is a cryptic membrane protein originally identified in *Shigella* and contributes to intestinal inflammation [16]. SepA is widespread among EAEC strains [16]. EAEC are highly heterogeneous in their virulence repertoire, and none of the above-mentioned virulence factors are found in all EAEC isolates and no single factor has ever been implicated in EAEC virulence.

Genetically *E. coli* strains are classified into five main phylogenetic classes, A, B1, B2, D, and F [21]. Extra-intestinal *E. coli* strains mostly belong to group B2 and, to a lesser extent, group D, whereas commensal strains belong to group A [21]. Among DEC, ETEC, in particular, is predominantly identified in phylogroups A and B1, while EPEC is most often identified in phylogroups B2 and B1 [21]. EAEC phylogeny is complex, and EAEC strains may be distributed across all phylogroups [21]. Phylogenetic classes vary in their genotypic and phenotypic characteristics, their profiles of antibiotic resistance and virulence genes [22].

Throughout India, EAEC remains the most prevalent pathotype isolated from diarrheal diseases below age 5 years [23]. In the Global Enteric Multicenter Study (GEMS), a prospective case-control study of moderate to severe diarrhea in 0-59 month old children living in Africa and Asia, the attributable fraction of EAEC to moderate and severe diarrhea in Bangladesh was 9.9% among 12-23-month-olds [24]. In another study of malnutrition and enteric disease study (MAL-ED), asymptomatic EAEC illnesses were common early in life, along with clinical growth shortfalls. EAEC infection interactions with intestinal inflammation were limited in scale, but imply a pathway for the growth impact [25]. In India, EAEC prevalence was 7.6% in children under 5 years of age at the National Institute of Cholera and Enteric Diseases (NICED), Kolkata [26]. An older south Indian study in Vellore, however, showed EAEC to be as common in cases and controls [27]. No studies are available in India on aEAEC, and the relative proportions of tEAEC and aEAEC strains contributing to diarrhea have not been well studied in the Indian context. Therefore, the present study was planned to establish the epidemiological significance of tEAEC and aEAEC in childhood diarrhea in North India and to identify and compare the virulence determinants, phylogroups profile, and antibiotic susceptibility of aEAEC in comparison with tEAEC.

## Methods

### Ethics statement

The study was approved by the Postgraduate Institute of Medical Education and Research (PGIMER) Ethics Committee (INT/IEC/2017/173). Written informed consent was obtained from the patient parent/guardian.

### Study site and patients

A total of 1210 stool samples from children (age<10 years) with acute diarrhea were obtained from PGIMER and its referral laboratories in different regions across North India during the period from 2015 to 17. Thirteen laboratories participated in the study from Chandigarh, Haryana, Punjab, Uttarakhand, and Himachal Pradesh (Fig 1). Diarrhea was defined as the passage of three or more liquid or semi-liquid stools. Clinical data of the diarrhoeal episodes were collected from each case. Vesikari score was used to assess an EAEC-associated episode of diarrhea [28]. Elements of the ranking include the length of diarrhea (in days; score, 0 to 3 points), the highest number of stools per day during the episode (score, 1 to 3 points), the occurrence of vomiting (score, 0 to 1 point), the maximum number of emeses per day during the episode (value, 0 to 3 points), the existence of fever (score, 0 to 1 point), the presence of fatigue (score, 0 to 1 point) and treaties. The highest score was 14. The control group included stool samples from 550 (age<5 years) stable anganwadi/school children obtained during the period of 2016–17 in the Chandigarh area.

### Sample collection and processing

Stool samples were collected in sterile containers, transferred to Cary Blair media, and transported to the laboratory in the cold chain. They were cultured for the presence of *V.cholerae*, *Aeromonas spp*., *Salmonella spp*., *Shigella spp*., and *E. coli* by standard procedures [29]. Briefly, samples were inoculated onto MacConkey agar, ampicillin blood agar, xylose lysine deoxycholate agar (XLD agar), thiosulfate-citrate-bile salts-sucrose agar (TCBS agar), alkaline peptone water (APW) and selenite F broth and incubated at 37˚C for 18–24h. Organisms were identified by standard biochemical [29], and Matrix-Assisted Laser Desorption/Ionization-Time Of Flight (MALDI-TOF), which was performed on a MALDI Microflex LT mass spectrometer (Bruker Daltonik GmbH, Bremen, Germany), and confirmed by serotyping using antisera

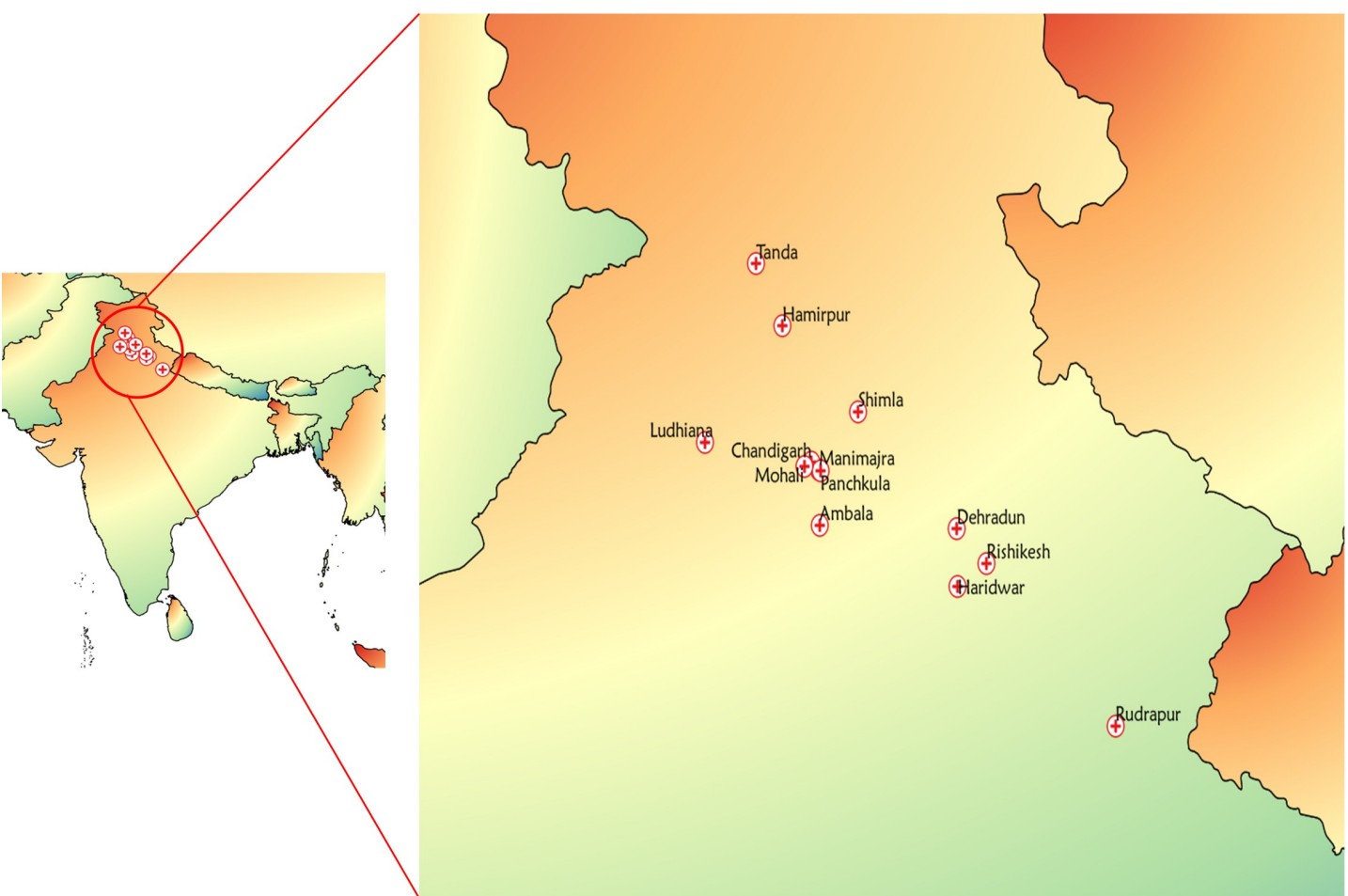

**Fig 1. Representation of sampling sites of the regional health centers of different cities across the North India region [Chandigarh (PGIMER), Civil hospital Manimajra; Haryana (Civil hospital Panchkula and Civil hospital Ambala Cantt); Himachal Pradesh (Government Medical College (GMC) Hamirpur, Indira Gandhi Medical College (IGMC) Shimla, and Rajendra Prasad Government Medical College (RPGMC) Tanda); Punjab (Civil hospital and private laboratory Mohali and Private laboratory Ludhiana); and Uttarakhand (All India Institute of Medical Science (AIIMS) Rishikesh, District hospital Rudrapur and District hospital Haridwar)] from where stool samples of diarrheal and healthy children were collected during 2015–17.** The map was made with Natural Earth using QGIS software version 3.14.6.

from Denka-Seiken (Japan). Up to three confirmed *E. coli*, colonies, were selected from each MacConkey agar plate and streaked onto fresh, sterilized nutrient agar and were stored in trypticase soy broth containing 15% glycerol at -80 °C.

## Extraction of DNA from confirmed *E. coli*

DNA from each confirmed *E. coli* isolates was extracted by the heat extraction method [30]. In 100μl deionized water, a single colony of biochemically confirmed *E. coli* isolates was emulsified and boiled for 5 min. The mixture was centrifuged at 10,000 g for 1 minute, and DNA containing supernatant was collected and stored at 4°C until further usage.

## Screening of DEC genes

Multiplex PCR for enterotoxins (heat-labile [LT] and heat-stable [ST]), EPEC (Eae) protein bundle forming protein (Bfp), Shiga toxins (Stx1, Stx2), VTcom for EHEC and (CVD432) for EAEC was performed using published primers and protocol as shown in Table 1 [31]. The

**Table 1. List of primers, their sequences, and the size of the amplified products, used in this study.**

| Target gene | Primers sequence | Primer designation | PCR product size (bp) | Reference |
|---|---|---|---|---|
| *bfp* | GGAAGTCAAATTCATGGGGGTAT GGAATCAGACGCAGACTGGTAGT | Bfp | 300 | [32] |
| *eae* | TCAATGCAGTTCCGTTATCAGTT GTAAAGTCCGTTACCCCAACCTG | Eae | 482 | [32] |
| *elt* | ACGGCGTTACTATCCTCTC TGGTCTCGGTCAGATATGTG | LT | 273 | [33] |
| *CVD432* | CTGGCGAAAGACTGTATCAT AATGTATAGAAATCCGCTGTT | pCVD432 | 630 | [34] |
| *estA1* | TCTTTCCCCTCTTTTAGTCAG ACAGGCAGGATTACAACAAAG | STp | 166 | [33] |
| *estA2-4* | TTCACCTTTCCCTCAGGATG CTATTCATGCTTTCAGGACCA | STh | 120 | [33] |
| *stx1* | CAGTTAATGTGGTGGCGAAGG CACCAGACAATGTAACCGCTG | Stx1 | 348 | [32] |
| *stx2* | ATCCTATTCCCGGGAGTTTACG GCGTCATCGTATACACAGGAGC | Stx2 | 584 | [32] |
| *stx1+stx2* | GAGCGAAATAATTTATATGTG TGATGATGGCAATTCAGTAT | VTcom | 518 | [35] |

pCVD432 primer utilized in this study for detection of EAEC, amplified the 630 bp region from start position 65 to the end position 694 of CVD432 gene.

## PCR conditions

Two or more sets of primers were used in multiplex PCR reactions. All PCR reactions were conducted at a final volume of 20μl comprising 0.5μl DNA, 1μl DNA polymerase, buffer, 2.6mM of each dNTP, 1.5 mM of $MgCl_2$, at a final concentration of 1.5mM. For the final PCR reaction, additional $MgCl_2$ to final 2 mM concentration and each primer to final 10μM concentration were used.

The thermocycling parameters for all PCRs were as follows. 95°C for 2 min, 95°C for 15s, 52°C for 8s, and 10s 72°C for 30 cycles, with a final 2 min extension at 72°C, and all PCRs were performed in the thermalocycler (Applied Biosystem Veriti 96 Well Thermal Cycler). Amplified samples were tested for 1.5% agarose gel electrophoresis in Tris–borate–EDTA and EtBr staining.

## Defining criteria for tEAEC and aEAEC by PCR

The tEAEC and aEAEC were identified using the *agg*R gene using primers, as described in Table 2. EAEC strains positive for *agg*R gene were classified as typical, and *agg*R-negative isolates were described as atypical EAEC. For PCR, the *agg*R gene was amplified to a total volume of 25 μl comprising 2.6 mM of each dNTP, 0.5 mM each primer, 10X PCR buffer, 1.5 mM $MgCl_2$, 1U Taq polymerase, and 1μl bacterial DNA. PCR parameters were as follows (1) 2 min denaturation at 95°C, (2) 50s denaturation at 94°C, (3) 1.5 min annealing, at 57 °C and (4) 1.5-min extension at 72°C with 35 cycles returning to step 2. The final extension at 72°C was for 10 minutes. Amplified products were studied using 1.5% agarose gel electrophoresis and visualized with ethidium bromide (EtBr) staining.

## Detection of virulence factors by PCR

*E. coli* strains confirmed as EAEC were further investigated for virulence genes. PCR for virulence genes, including *aap*, *pet*, *sig*A, *pic*, *sep*A, *sat*, *aai*C, *agg*4A, *aaf*A, *ast*A, *sep*A, *sat*, ORF3, *agg*A, *agg*3A, *aaf*C, ORF61, *eil*A, *cap*U, *air*, *esp*Y2, and *rmo*A was performed separately in a

**Table 2. Primers used for the 4 multiplex polymerase chain reactions (M-PCRs) and 3 monoplex PCRs, target gene description, base-pair size, annealing temperature, and primers concentration.**

| Multiplex PCR | Gene/ Target | Description of Target | Primer Sequence (5'- 3') | PCR Product, bp | Annealing Temperature Primer Concentration (_C), pmol/lL | GenBank Accession No. | References |
|---|---|---|---|---|---|---|---|
| M-PCR-1 | astA | EAST-1 heat-stable toxin | ATGCCATCAACACAGTAT GCGAGTGACGGCTTTGTAGT | 110 | 58/20 | L11241 | [36] |
| | pet | Plasmid-encoded toxin | GGCACAGAATAAAGGGGTGTTT CCTCTTGTTTCCACGACATAC | 302 | 58/25 | AF056581 | [37] |
| | sigA | IgA protease-like homolog | CCGACTTCTCACTTTCTCCCG CCATCCAGCTGCATAGTGTTTG | 430 | 58/30 | NC_004337 | [16] |
| | pic | Serine protease precursor | ACTGGATCTTAAGGCTCAGGAT GACTTAATGTCACTGTTCAGCG | 572 | 58/25 | AF097644 | [37] |
| | sepA | Shigella extracellular protease | GCAGTGGAAATATGATGCGGC TTGTTCAGATCGGAGAAGAACG | 794 | 58/25 | Z48219 | [37] |
| | sat | Secreted autotransporter toxin | TCAGAAGCTCAGCGAATCATTG CCATTATCACCAGTAAAACGCACC | 932 | 58/25 | AE014075 | [16] |
| M-PCR-2 | ORF3 | Cryptic protein | CAGCAACCATCGCATTTCTA CGCATCTTTCAATACCTCCA | 121 | 57/35 | AB261016.2 | [16] |
| | aap | Dispersin, protein | GGACCCGTCCCAATGTATAA CCATTCGGTTAGAGCACGAT | 250 | 57/25 | Z32523 | [16] |
| | aaiC | AaiC, secreted protein | TGGTGACTACTTTGATGGACATTGT GACACTCTCTTCTGGGGTAAACGA | 313 | 57/25 | AB255435.1 | [16] |
| | aggR | Transcriptional activator | GCAATCAGATTAARCAGCGATACA CATTCTTGATTGCATAAGGATCTGG | 426 | 57/25 | Z18751 | [16] |
| M-PCR-3 | agg4A | AAF/IV fimbrial subunit | TGAGTTGTGGGGCTAYCTGGA CACCATAAGCCGCCAAATAAGC | 169 | 57/25 | EU637023 | [16] |
| | aggA | AAF/I fimbrial subunit | TCTATCTRGGGGGGCTAACGCT ACCTGTTCCCCATAACCAGACC | 220 | 57/25 | Y18149 | [16] |
| | aafA | AAF/II fimbrial subunit | CTACTTTATTATCAAGTGGAGCCGCTA GGAGAGGCCAGAGTGAATCCTG | 289 | 57/25 | AY344586 | [16] |
| | agg3A | AAF/III fimbrial subunit | CCAGTTATTACAGGGTAACAAGGGAA TTGGTCTGGAATAACAACTTGAACG | 370 | 57/25 | AF411067 | [16] |
| | aafC | Usher, AAF/II assembly unit | ACAGCCTGCGGTCAAAAGC GCTTACGGGTACGAGTTTTACGG | 491 | 57/25 | AF114828 | [16] |
| M-PCR-4 | ORF61 | Plasmid-encoded hemolysin | AGCTCTGGAAACTGGCCTCT AACCGTCCTGATTTCTGCTT | 108 | 57/25 | J02459.1 | [16] |
| | eilA | Salmonella HilA homolog | AGGTCTGGAGCGCGAGTGTT GTAAAACGGTATCCACGACC | 248 | 57/25 | CP009685.1 | [16] |
| | capU | Hexosyltransferase homolog | CAGGCTGTTGCTCAAATGAA GTTCGACATCCTTCCTGCTC | 395 | 57/25 | AF134403 | [16] |
| | air | Enteroaggregative immunoglobulin repeat protein | TTATCCTGGTCTGTCTCAAT GGTTAAATCGCTGGTTTCTT | 600 | 57/25 | CP009685.1 | [16] |
| Monoplex PCR | espY2 | Non-LEE-encoded type III secreted effector | CGCAAAAGATCCGGAAAATA TCAGCATTGCTCAGGTCAAC | 216 | 59/25 | ECSP_0073 | [16] |
| Monoplex PCR | rmoA | Putative hemolysin expression- modulating protein | TTACCTTACATATTTCCATATC CGAAAACAAAACAGGAATGG | 210 | 60/25 | ECUMN_0072 | [16] |
| Monoplex PCR | shiA | shiA-like inflammation suppressor | CAGAATGCCCCGCGTAAGGC CACTGAAGGCTCGCTCATGATCGCCG | 292 | 57/25 | ECB_03517 | [38] |

25μl reaction mixture containing 2.5 μl 10X PCR buffer, 1 mM MgCl$_2$, 1 mM each dNTP, 0.5 U *Taq* DNA polymerase and 5 μl DNA. The target primer sequences, concentrations, annealing temperatures, and PCR product sizes for different virulence genes are described in Table 2.

### Phylogenetic analysis via PCR

A triplex PCR was used to detect phylogenetic groups A, B1, B2, and D by amplifying the following gene targets. *chu*A, *yja*A, and a cryptic DNA fragment, TspE4C2 [39]. The classification was correlated with Clermont dichotomous decision tree [40].

### Antibiotic sensitivity testing

Antibiotic sensitivity testing (AST) was performed with the disc diffusion method for the following antibiotics: Ampicillin (10μg), ciprofloxacin (5μg), amikacin (30μg), Imipenem (10μg), levofloxacin (5 μg), gentamicin (10μg), cefepime (15μg), piperacillin-tazobactam (10μg), ertapenem (10μg), cotrimoxazole (25μg), cefoxitin (30μg), and ceftriaxone (30μg) according to CLSI guidelines [41]. Multidrug resistance (MDR) was defined as an acquired resistance toward three or more antibiotics from different antibiotic classes tested.

### Statistical analysis

A two-tailed chi-square test was used to compare groups. If low predicted values constrained the study, Fisher's exact test was used. Odds ratio (OR) and 95% confidence intervals (CIs) were calculated using the GraphPad PRISM program.

## Results

### Children with acute diarrhea (diarrheal group)

From 1210 children with acute diarrhea, a total of 273 (22.5%) DEC were detected by M-PCR. The following DEC pathotypes were identified: EAEC 11.4% (138/273), EPEC 6.1% (75/273), and ETEC 4.9% (60/273). Another three (3/273, 0.25%) children had an infection with hybrid strains (2 EAEC/EHEC, 1 EPEC/ETEC), i.e., *E. coli* strains carrying defining genes for more than one DEC as shown in Fig 2.

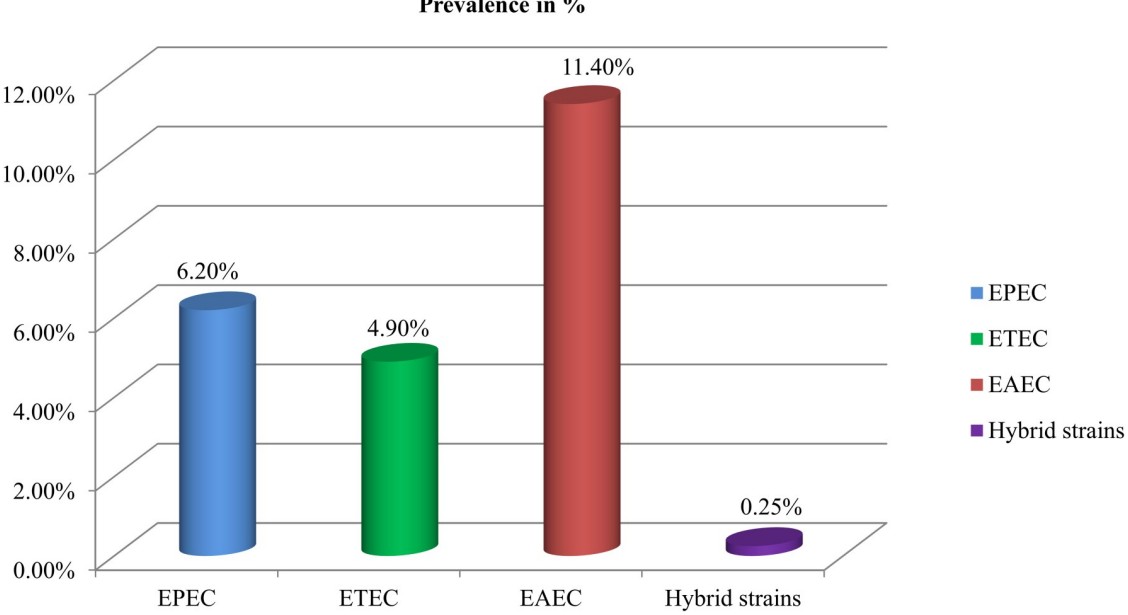

**Fig 2. Distribution of diarrheagenic *Escherichia coli* (DEC) groups in the diarrheal group.** EPEC: enteropathogenic *E. coli*, EAEC: enteroaggregative *E. coli*, ETEC: enterotoxigenic *E. coli* and hybrid strains.

**Table 3.** Distribution of DEC groups by age in children with acute diarrhea.

| | Number of children | Children' age in years (%) | | | | P- value |
|---|---|---|---|---|---|---|
| | | 0–1 | 1–2 | 2–5 | 5–10 | |
| | 1210 | 184 (15.2) | 130 (10.7) | 383 (31.6) | 513 (42.3) | |
| Total DEC | 273 (22.5) | 45 (24.4) | 48 (36.9) | 106 (28) | 84 (16.3) | 0.0001* |
| EAEC infection | 138 (11.4) | 26 (14.1) | 20 (15.3) | 50 (13.05) | 42 (8.18) | 0.002* |
| EPEC infection | 75 (6.1) | 15 (8.15) | 8 (6.15) | 35 (9.1) | 17 (3.3) | 0.0003* |
| ETEC infection | 60 (4.9) | 6 (3.2) | 8 (6.1) | 21 (5.4) | 25 (4.8) | 1.0 |
| Hybrid DEC infection | 3 (0.24) | 1 (0.5) | 0 (0) | 2 (0.5) | 0 (0) | 0.2 |

*Statistically significant (p<0.05) when compared pooled prevalence of EAEC in children aged 0–5 years to children aged 5–10 years. Fischer's exact test was used to compare the presence of DEC in children with diarrhea at different age groups

ETEC- Enterotoxigenic *E.coli*

EAEC-Enteroaggregative *E. coli*

EPEC- Enteropathogenic *E. coli*

Hybrid DEC -*E. coli* strains carrying defining genes for more than one DEC

The prevalence of DEC was higher in children less than 5 years, as shown in Table 3. Besides, 16.3% (84/273) of DECs occurred in children between 5–10 years. Among DEC, 11.4% (138/1210) of children had an infection with EAEC as compared to EPEC 6.1% (75/1210) and ETEC 4.9% (60/1210) as shown in Table 3 and Fig 2. The gel electrophoresis profile of different DEC using M-PCR was shown in Fig 3. Following statistically significant correlations of age and illness with a particular DEC category were observed. EAEC diarrhea was found more often in children aged 0–5 than children aged 5–10 years (p = 0.002) shown in Table 3. Similarly, a statistically significant correlation between age and prevalence of DEC and EPEC (p = 0.0001 and p = 0.0003, respectively) was found (Table 3).

## Prevalence of DEC pathotypes among 550 children without diarrhea (control group)

Among 550 healthy children from the community having no diarrhea, a total of 77 (14%) DEC were detected via M-PCR. The following DEC pathotypes were identified. EAEC 6% (33/77), EPEC 4% (21/77), and ETEC 4.18% (23/77) (Table 4). In the control group, there were statistically significant associations between age and infection with a specific DEC group as follows; DEC infection was more frequently observed among children belonging to 0–2 years age group than children in 2–5 years of age group (p = 0.001) shown in Table 4. Also, there was statistically significant associations of EPEC infection with age group 0–2 vs. 2–5 years of age (p = 0.02). Overall, DEC and EPEC in healthy children is significantly associated in 0–2 years. For others, there was no significant difference (Table 4).

## Distribution of DEC categories among diarrheal and healthy children

The prevalence of DEC was observed to be 22.5% in cases of acute diarrheal children, whereas in healthy children, it was 14%. Among DEC pathotypes, EAEC was the most prevalent pathotypes accounting for 11.4% in diarrheal children and 6% in healthy children, as shown in Table 5. The overall prevalence of DEC was statistically significant when compared with the control or healthy individuals (p = 0.0001) (Table 5). Also, the occurrence of EAEC and EPEC was statistically significant among the two groups (Table 5).

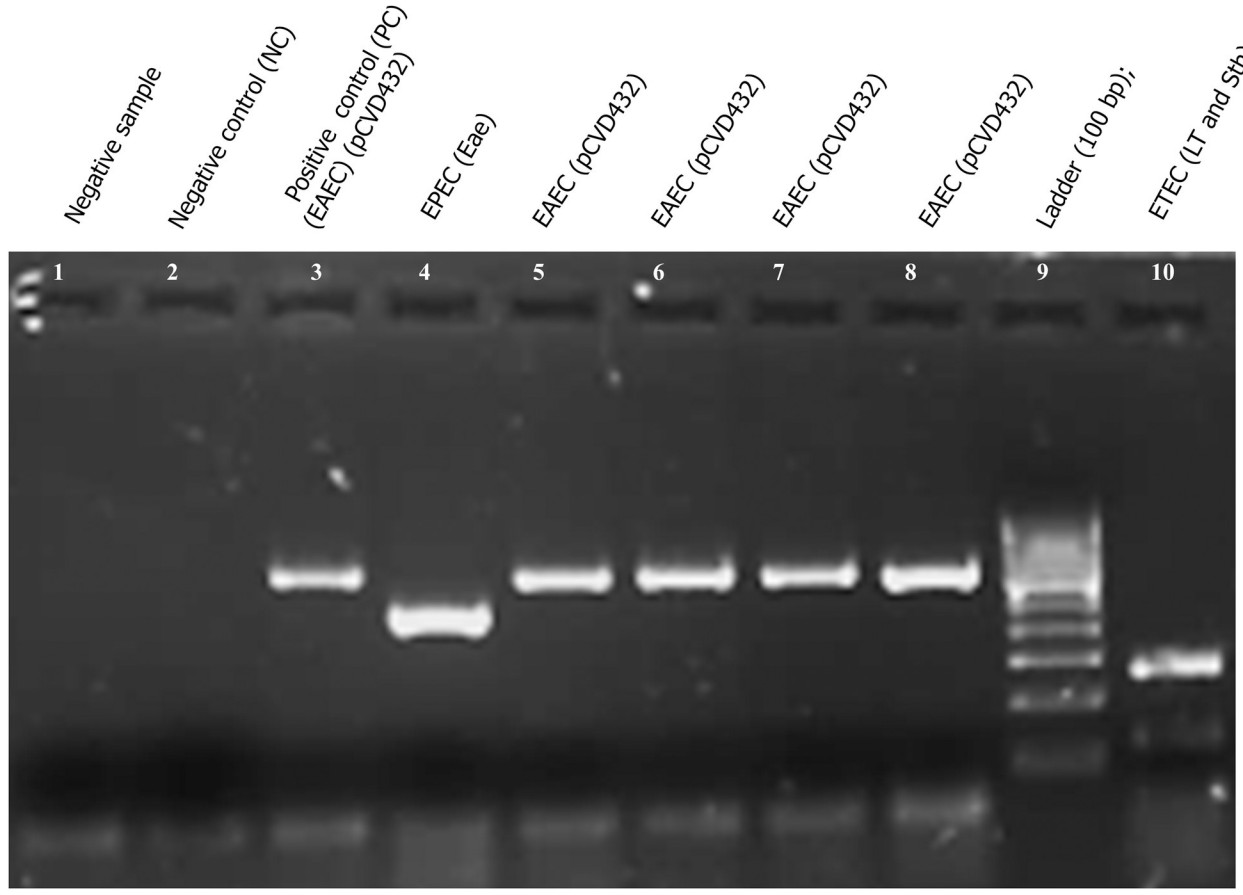

**Fig 3. A representative gel electrophoresis profile of different DEC using M-PCR.** lane 1, negative sample; lane 2, negative control (NC); lane 3, positive control (PC) EAEC (pCVD432); lane 4, EPEC (Eae); lane 5, EAEC (pCVD432); lane 6, EAEC (pCVD432); lane 7 EAEC (pCVD432); lane 8, EAEC (pCVD432); lane 9, ladder (100 bp); and lane 10, ETEC (LT and Sth).

**Table 4. Distribution of DEC groups by age in children without diarrhea.**

|  | Number of children (%) | Children age in years (%) | | | P-value |
|---|---|---|---|---|---|
|  |  | 0–1 | 1–2 | 2–5 |  |
|  | 550 | 109 (19.8) | 149 (27) | 292 (53) |  |
| Total DEC | 77 (14) | 21 (19.2) | 25 (32.4) | 31 (10.6) | 0.001* |
| EAEC | 33 (6.0) | 9 (8.2) | 10 (6.7) | 14 (4.79) | 0.2 |
| EPEC | 21 (4.0) | 5 (4.5) | 10 (6.7) | 6 (2.05) | 0.02* |
| ETEC | 23 (4.18) | 7 (6.4) | 6 (4.02) | 10 (3.42) | 0.3 |
| Hybrid DEC | 0 (0) | 0 (0) | 0 (0) | 0 (0) |  |

*Statistically significant (p<0.05) when compared pooled prevalence of EAEC in children 0–2 years to children aged 2–5 years. Fischer's exact test was used to compare the presence of DEC in children without diarrhea at different age groups

ETEC- Enterotoxigenic *E.coli*

EAEC- Enteroaggregative *E. coli*

EPEC- Enteropathogenic *E. coli*

Hybrid DEC -*E. coli* strains carrying defining genes for more than one DEC

**Table 5. Distribution of DEC pathotypes among diarrheal and healthy children.**

| DEC category | No. of strains (%) | | |
|---|---|---|---|
| | Diarrheal sample n (%) | Control samples n (%) | *P*-value |
| Total DEC | 273 | 77 | 0.0001* |
| EAEC | 138 (50.5) | 33 (42.8) | 0.0003* |
| ETEC | 60 (21.9) | 23 (29.8) | 0.4 |
| EPEC | 75 (27.4) | 21 (27.2) | 0.04* |

*Statistically significant (P<0.05). Data was analysed by using Fischer's exact test.

## Prevalence of other bacterial enteropathogens among 1210 children with acute diarrhea

Other bacterial enteropathogens like *Vibrio cholerae* (6.8%), *Aeromonas* (2.97%), *Salmonella* (0.9%), and *Shigella* (3.08%), *Bacillus cereus* (0.89%) were also detected from children with diarrhea.

## Severity score

S1 Table (supplementary data) represents the region-wise severity score of 1210 stool samples collected from children with acute diarrhea. Most of the patients presented with severe diarrhea (48.1%) and moderate diarrhea (30%). When severe and moderate severity was pooled and compared with mild severity, most regions showed statistically significant differences among diarrheal children (S1 Table). However, in most regions, severity scores did not vary a lot (S1 Table of supplementary data).

The severity score of cases from whom EAEC was isolated is shown in Table 6. Severity among EAEC infected children was as follows; severe 39 (28.6%); mild 53 (38.4%); moderate 46 (33.3%). Correlation of severity with age found that children belonging to 0–5 years of age group were more (69.5%) affected in comparison to the children 5–10 years of age (30.4%) (Table 6).

## The prevalence of typical and atypical EAEC

All 171 EAEC strains from healthy (33) and acute diarrheal children (138) were tested for the presence of the *agg*R gene, 91(66%) of 138 were classified as tEAEC, and 47 (34%) as aEAEC strains (p = 0.0001) (Table 7). In the control group, 11 (33%) strains out of 33 isolates were typical, and 22 (66%) were atypical. aEAEC strains were more common in the control group (66%) than the diarrheal group (34%) (p = 0.01), as shown in Table 7.

**Table 6. The severity of EAEC infected children based on the number, age, and gender.**

| Severity | No. of EAEC detected (%) | Sex-wise distribution | Age-wise distribution | |
|---|---|---|---|---|
| Severe | 39 (28.6%) | Male = 24 Female = 15 | 0–2 yrs. | Severe = 12 Mild = 19 Moderate = 15 |
| Mild | 53 (38.4%) | Male = 26 Female = 27 | 2-5yrs. | Severe = 15 Mild = 20 Moderate = 15 |
| Moderate | 46 (33.3%) | Male = 34 Female = 12 | 5–10 yrs. | Severe = 12 Mild = 14 Moderate = 16 |

**Table 7. Distribution of tEAEC and aEAEC isolates in diarrheal and control group.**

| Diarrheal group | | | *P*-value | Control group | | | *P*-value |
|---|---|---|---|---|---|---|---|
| Total EAEC | tEAEC | aEAEC | | Total EAEC | tEAEC | aEAEC | |
| 138 | 91 (66%) | 47 (34%) | 0.0001* | 33 | 11 (33%) | 22 (66%) | 0.01* |

*Statistically significant (p<0.05) when tEAEC and aEAEC were compared in either diarrheal and control group. Data was analysed by using Fischer's exact test.

### Frequencies of virulence-related genes among EAEC strains in cases and controls

To evaluate the functions of putative virulence factors in EAEC epidemiology, 4 M-PCR assays were used to define 21 virulence genes in EAEC strains. S2 Table (supplementary data) shows PCR results for all strains. Of the 21 genes identified, the most prevalent was the *ast*A (87.6%) in the diarrheal group followed by ORF3 (71.8%), ORF61 (69.5%), *aap* (61.6%) and *cap*U (52.8%) in EAEC strains respectively (S2 Table of Supplementary data). The most prevalent AAF pilin gene was AAF/IV (*agg*4A) (43.6%) followed by AAF/I encoded by (*agg*A) (24.6%), AAF/III encoded by *agg*3A (5.07%) and AAF/II (*aaf*A) (3.6%) in cases (S2 Table of Supplementary data). Of the 5 genes encoding SPATEs, the most common among the diarrheal group were *sat* (14.5%), *sep*A (10.14%), and *pic* (13.0%). The least frequent SPATEs were *pet* (7.24%) and *sig*A (2.8%), as shown in (S2 Table of Supplementary data).

In the control group, *ast*A (90%) was the most prevalent gene, followed by ORF61 (66.6%) and ORF3 (54.5%). The most prevalent adhesive variant was *agg*A (12.1%) and *agg*4A (9.0%). Of the 5 genes encoding SPATEs, the most common were *pet* (30.3%), *sat* (9.0%), and *pic* (9.0%), and the least common SPATEs was *sig*A (6.0%).

Statistically, there was a significant difference in *pet* (p = 0.0006), ORF3 (p = 0.05), *agg*4A (p = 0.0012) and *esp*Y (p = 0.01) from the diarrheal and control group (S2 Table of Supplementary data).

### Prevalence of tEAEC and aEAEC and its virulence-related genes in the diarrheal group

The distribution of virulence markers in tEAEC from the diarrheal samples was as follows; hypothetical ORF3 was the most frequently detected (98.9%) followed by *ast*A (95.6%), *aap* (89%), ORF61 (75.8%) and *cap*U (59.3%) in tEAEC strains respectively (Table 8 and S1 Fig). In aEAEC isolates, on the other side, *ast*A (72.3%) was the most prevalent gene among 21 virulence genes. The most frequent adhesive variant was that of AAF/III, encoded by *agg*3A (54.9%), followed by those of AAF/IV (*agg*4A) (47.2%), AAF/I (*agg*A) (29.6%), and AAF/II (*aaf*A) (3.2%) in tEAEC strains (Table 8). Whereas among aEAEC strains, the most prevalent adhesive gene was *agg*A (14.8%) and *aaf*C (14.8%), respectively. Of the 5 genes encoding SPATEs, the most frequent were *sat* (15.3%), *sep*A (13.1), and *pic* (13.0%). The least common SPATEs was *sig*A (2.8%) in tEAEC strains, as shown in (Table 8). Whereas in aEAEC isolates, most detected SPATEs genes were *pic* (14.8%) and *sat* (12.7%), respectively (Table 8). Statistically, a significant difference was observed for the following virulence genes *ast*A (p = 0.0002), ORF3 (p = 0.001), *aap* (p = 0.0001), *aai*C (0.003), *aaf*C (0.04), ORF61 (0.03), *cap*U (0.04) and *esp*Y (p = 0.01) among tEAEC and aEAEC in diarrheal group (Table 8 and S1 Fig).

### Prevalence of tEAEC and aEAEC and its virulence-related genes in the control group

Of the 21 genes scored, among tEAEC in the control group, *ast*A was the most frequently detected (100%), followed by *aap* (90.9%), ORF3 (81.8%), and ORF61 (63.6%) whereas in

**Table 8. Distribution of tEAEC and aEAEC virulence-related markers in the diarrheal group.**

| EAEC factor | Diarrheal group | | | | Control group | | | |
|---|---|---|---|---|---|---|---|---|
| | Total (%) n = 138 | tEAEC n = 91 (%) | aEAEC n = 47 (%) | *P* value | Total (%) n = 33 | tEAEC n = 11(%) | aEAEC n = 22 (%) | *P*-value |
| *ast*A | 121 (87.6) | 87 (95.6) | 34 (72.3) | 0.0002* | 30 (90.9) | 11 (100) | 19 (86.36) | 0.5 |
| *sig*A | 4 (2.8) | 2 (2.2) | 2 (4.2) | 0.6 | 2 (6.0) | 0 (0) | 2 (9.0) | 0.5 |
| *Pic* | 18 (13.0) | 11 (12.0) | 7 (14.8) | 0.7 | 3 (9.0) | 1 (9.0) | 2 (9.0) | 1.0 |
| *sep*A | 14 (10.14) | 12 (13.1) | 2 (4.2) | 0.1 | 3 (9.0) | 2 (18.0) | 1 (4.5) | 0.2 |
| *sat* | 20 (14.5) | 14 (15.3) | 6 (12.7) | 0.8 | 3 (9.0) | 3 (27.0) | 0 (0) | 0.03* |
| *pet* | 10 (7.24) | 6 (6.5) | 4 (8.5) | 0.7 | 10 (30.3) | 3 (27.0) | 7 (31.8) | 1.0 |
| ORF3 | 99 (71.8) | 90 (98.9) | 9 (19.1) | 0.0001* | 18 (54.5) | 9 (81.8) | 9 (40.9) | 0.03* |
| *aap* | 85 (61.6) | 81 (89.0) | 4 (8.5) | 0.0001* | 16 (48.4) | 10 (90.9) | 6 (27.2) | 0.02* |
| *aai*C | 22 (16.4) | 21 (23.0) | 1 (2.1) | 0.003* | 8 (24.2) | 4 (36.0) | 4 (18.1) | 0.3 |
| *agg*4A | 60 (43.4) | 43 (47.2) | 17 (36.1) | 0.2 | 3 (9.0) | 1 (9.0) | 2 (9.0) | 1.0 |
| *agg*A | 34 (24.6) | 27 (29.6) | 7 (14.8) | 0.06 | 4 (12.1) | 0 (0) | 4 (18.1) | 0.2 |
| *aaf*A | 5 (3.6) | 3 (3.2) | 2 (4.2) | 1.0 | 2 (6.0) | 0 (0) | 2 (9.0) | 0.5 |
| *agg*3A | 07 (5.07) | 5 (54.9) | 2 (4.2) | 1.0 | 1 (3.0) | 0 (0) | 1 (4.5) | 1.0 |
| *aaf*C | 11 (7.9) | 4 (4.3) | 7 (14.8) | 0.04* | 1 (3.0) | 1 (11) | 0 (0) | 0.3 |
| ORF61 | 96 (69.5) | 69 (75.8) | 27 (57.4) | 0.03* | 22 (66.6) | 7 (63.6) | 15 (68.1) | 1.0 |
| *eil*A | 57 (41.3) | 39 (42.8) | 18 (38.2) | 0.7 | 15 (45.4) | 6 (54.5) | 9 (40.9) | 0.4 |
| *cap*U | 73 (52.8) | 54 (59.3) | 19 (40.4) | 0.04* | 15 (45.4) | 6 (54.5) | 9 (40.9) | 0.4 |
| *esp*Y | 50 (36.2) | 39 (42.8) | 11 (23.4) | 0.02* | 4 (12.12) | 3 (27.0) | 1 (4.5) | 0.09 |
| *rmo*A | 62 (44.9) | 37 (40.6) | 25 (53.1) | 0.2 | 15 (45.4) | 5 (45.4) | 10 (45.5) | 1.0 |
| *shi*A | 30 (21.7) | 20 (21.9) | 10 (21.2) | 1.0 | 9 (27.2) | 3 (27.2) | 6 (27.2) | 1.0 |
| *air* | 28 (20.) | 20 (21.9) | 8 (17.0) | 0.6 | 4 (12.12) | 1 (9.0) | 3 (13.6) | 1.0 |

*Statistically significant (P<0.05) when tEAEC and aEAEC strains from the diarrheal group and control group were compared. Data was analysed by using Fischer's exact test.

aEAEC isolates *ast*A was detected in 86.36% isolates (Table 8 and S1 Fig). The distribution of adhesive genes was most prevalent in the aEAEC strains as compare to tEAEC, as shown in Table 8. Of the 5 genes encoding SPATEs, the most prominent were *pe*t (27%) and *sat* (18%) in tEAEC as compared to *pet* (31.8%) and *pic* (9.0%) in aEAEC isolates (Table 8). Among all the putative virulence factor scored, as *sat* (p = 0.03), ORF3 (p = 0.003), *aap* (p = 0.02), were significantly associated among tEAEC and aEAEC isolates in the control group shown in Table 8 and S1 Fig.

Based on virulence genes analysis in EAEC isolates among diarrheal and control samples, several different combinations of virulence markers were found among the EAEC isolates in our study (S3 Table of supplementary data). Gene combination *ast*A, ORF3, *aap*, *agg*R, ORF61, *cap*U was most prominent among cases (26.8%) and controls (9.09%), respectively (S3 Table of supplementary data).

## Distribution of phylogenetic groups among EAEC isolates from cases and controls

Among EAEC isolates, phylogenetic group D (43.4%) and B1 (39.39%) were the most prevalent in diarrheal and control groups, respectively. Phylogenetic distribution in the case of diarrheal EAEC was as follows; D (43.4%), B1 (24.6%), B2 (23.1), and A (8.6%). Similarly, in the control group distribution of phylogroup was as follows; B1 (39.39%), D (30.3%), A (15.15%) and B2 (12.1%) (Table 9).

**Table 9. Phylogenetic distribution of tEAEC and aEAEC in the control group.**

| Phylogroups | Diarrheal group | | | | Control group | | | |
|---|---|---|---|---|---|---|---|---|
| | Total (%) n = 138 | tEAEC (n = 91) | aEAEC (n = 47) | *P*-value | Total (%) n = 33 | tEAEC (n = 11) | aEAEC (n = 22) | *P*-value |
| A | 12 (8.6) | 7 (7.6) | 5 (10.6) | 0.5 | 5 (15.15) | 0 (0) | 5 (22.7) | 0.1 |
| B1 | 34 (24.6) | 22 (24.1) | 12 (25.5) | 1.0 | 13 (39.39) | 4 (36.3) | 9 (40.9) | 1.0 |
| B2 | 32 (23.1) | 22 (24.1) | 10 (21.2) | 0.8 | 4 (12.1) | 2 (18.1) | 2 (9.0) | 0.5 |
| D | 60 (43.4) | 40 (43.9) | 20 (42.5) | 1.0 | 10 (30.3) | 4 (36.3) | 6 (27.2) | 0.6 |

*Statistically significant (P<0.05). When tEAEC and aEAEC strains from the diarrheal group and control group were compared. Data was analysed by using Fischer's exact test.

## Distribution of phylogenetic groups among tEAEC and aEAEC isolates from cases and controls

Phylogenetic distribution of tEAEC and atypical aEAEC in the diarrheal and control group is shown in Table 9. On comparing the phylogenetic distribution of tEAEC isolated from both diarrheal and control groups, phylogroup D was the most common (43.9% vs. 36.3%) followed by phylogroup B1 (24.1% vs. 36.3%) and phylogroup B2 (24.1% vs. 18.1%) (Table 9). Phylogroup A (8.6% vs. 0.0%) was the least common phylogroup. However, there was no significant association in tEAEC and aEAEC among diarrheal and control groups (Table 9).

## Antimicrobial susceptibility of tEAEC and aEAEC strains isolated from diarrheal and control group

All strains were found to be resistant to at least one of the antibiotics tested. The highest antibiotic resistance was observed in ampicillin (86%), cotrimoxazole (70.2%), ciprofloxacin (67.3%), ceftriaxone (63%), and levofloxacin (52%) in EAEC isolates from the diarrheal group (Table 10). The tEAEC isolates showed higher antibiotic resistance to ampicillin, cotrimoxazole, and ciprofloxacin. Resistance to ertapenem and amikacin were the lowest in both tEAEC

**Table 10. Antibiotic resistance in tEAEC and aEAEC strains isolated from the diarrheal and control group.**

| Antibiotics | Diarrheal group | | | | Control group | | | |
|---|---|---|---|---|---|---|---|---|
| | Total n = 138 (%) | tEAEC n = 91 (%) | aEAEC (n = 47) (%) | *P*-value | Total n = 33 (%) | tEAEC n = 11 (%) | aEAEC n = 22 (%) | *P*-value |
| Ampicillin | 118 (86) | 80 (87.9) | 38 (80.8) | 0.3 | 30 (90.9) | 11 (100) | 19 (86.3) | 0.5 |
| Ciprofloxacin | 92 (67.3) | 64 (70.3) | 28 (59.5) | 0.2 | 20 (60.6) | 7 (63.6) | 13 (59) | 1.0 |
| Amikacin | 7 (5.0) | 7 (7.6) | 0 (0) | 0.09 | 0 (0) | 0 (0) | 0 (0) | 1.0 |
| Imipenem | 13 (10.1) | 9 (9.8) | 4 (8.5) | 1.0 | 0 (0) | 0 (0) | 0 (0) | 1.0 |
| Levofloxacin | 72 (52.1) | 51 (56) | 21 (44.6) | 0.2 | 14 (42.4) | 5 (45.5) | 9 (40.9) | 1.00 |
| Gentamicin | 25 (18.1) | 15 (16.4) | 10 (21.2) | 0.4 | 0 (0) | 0 (0) | 0 (0) | 1.0 |
| Cefixime | 26 (19) | 14 (15.3) | 12 (25.5) | 0.17 | 6 (18.1) | 2 (18.1) | 4 (14.8) | 1.0 |
| PiperacillinTazobactam | 13 (9.4) | 8 (8.7) | 5 (10.6) | 0.7 | 1 (3.0) | 1 (9.0) | 0 (0) | 0.3 |
| Ertapenem | 3 (2.1) | 3 (3.2) | 0 (0) | 0.5 | 0 (0) | 0 (0) | 0 (0) | 1 |
| Cotrimoxazole | 97 (70.2) | 65 (71.4) | 32 (68) | 0.6 | 21 (63.6) | 8 (72.7) | 13 (59.0) | 0.2 |
| Cefoxitin | 42 (30.4) | 30 (32.9) | 12 (25.5) | 0.4 | 14 (42.4) | 5 (45.4) | 9 (40.9) | 1.0 |
| Ceftriaxone | 87 (63) | 67 (73.4) | 30 (63.8) | 0.2 | 17 (51.5) | 6 (54.4) | 11 (50.0) | 1.0 |

*Statistically significant (P<0.05). When tEAEC and aEAEC strains from the diarrheal group and control group were compared. Data was analysed by using Fischer's exact test.

and aEAEC in the diarrheal and control group (Table 10). MDR was observed in 75.8% of the EAEC strains. We have not found significant differences between tEAEC and aEAEC in cases and control (Table 10).

## Discussion

EAEC is a recognized cause of diarrhea in developed and industrialized nations, both in children and adults [6]. It is a key agent of traveler diarrhea, particularly travelers to developed countries like India, Mexico, and Jamaica [11]. Studies from developing countries like India, Brazil, Congo, Southwest Nigeria, have reported EAEC strains as important emerging agents of pediatric diarrhea [11, 42]. EAEC infections usually cause sporadic diarrhea but can also cause outbreaks [43–45]. While EAEC is an significant aetiological agent of diarrhea, pathogenic EAEC identification remains difficult. Furthermore, EAEC is known to cause asymptomatic colonization [46]. At the molecular level, a plethora of virulence factors and virulence-associated factors have been associated with clinically relevant isolates [47]. Nevertheless, a worldwide applicable marker has not been recognized for diarrheogenic EAEC detection, possibly reflecting geographical variability or differences among the populations studied, but also highlighting the heterogeneous character of EAEC.

There are geographic variations in the prevalence of EAEC. In our study, EAEC was present in a higher proportion (11.4%), followed by EPEC (6.2%) and ETEC (4.9%). In a recent study on diarrheal disease in children from Lima, Peru, EAEC was identified as the most frequent DEC (15.1%) led by EPEC (7.6%) and DAEC (4.6%), respectively [48]. Studies in Britain, Germany, USA, and Romania included EAEC as a prevalent bacterial source of diarrhea varying from 2% to 6% [49–52]. In Nepal, Brazil, and Mali, the rate of EAEC among diarrhea cases ranged from 4.5 to 39% [16, 53]. Studies from various parts of India reported variable prevalence (7.0% to 16.0%) of EAEC [54]. Kahali et al. (2004), in their hospital-based surveillance analysis from Kolkata, reported 6.6% EAEC prevalence and a prevalence of 5.12% in children <5 years of age [55]. Another NICED (India) research reported a 12% prevalence in children under 5 years of age [26]. All these studies have one common aspect that EAEC as a group was the most frequently identified pathogen in diarrheal children. However, EAEC is also isolated frequently from healthy children. In a study from Dhaka (Bangladesh) from 1993 to 1994, a high rate of healthy carriage of EAEC (7%) was reported in young children [56]. A recent study on EAEC from Mali in children aged 0 to 59 months revealed that almost 50% (61/121) of EAEC were isolated from the patients without diarrhea [16]. Similarly, in a study from South India, EAEC was detected in 9% of non-diarrheal individuals [57]. Therefore, reliable identification of EAEC strains from carrier strains may be extremely beneficial for effective management. We observed a 6.0% faecal carriage in healthy children. This is higher than the rates of 1.7% and 2% described in two studies between 2006 and 2009 from the USA with asymptomatic volunteers and outpatients, respectively [58, 59]. A potential reason for these geographical differences involves differences in socio-economic factors, laboratory facilities, and the environment.

The severity of EAEC with diarrhea tends to differ geographically [13, 60]. Most infected children in our study had moderate to severe forms of diarrhea. Clinical assessment of diarrheal illness showed (48.1%) patients had a severe illness, while (30%) patients presented with moderate illness. In our study, males predominated over females. Our results were in accordance with Ochoa et al. (2009), where they found 39.1% of severe and moderate diarrheal episodes due to EAEC infection [48]. In another study by Vilchej et al., (2009), identified 32.4% of EAEC in severe cases [61]. We also studied the correlation of severity with age. We found

that children belonging to 0–5 years of age group were more severely (31%) affected in comparison to the children 5–10 years of age (26%).

Our analysis showed a significant association between EAEC and age. We examined that 0–5 year-old children are highly susceptible to EAEC infection as opposed to the higher-age group. Children under the age of 5 are more susceptible to EAEC illness due to maternally obtained passive immunity and unhygienic incorporation of weaning food [62]. Crawling and teething are two processes which usually start at the age of 6 months and last up to 12 months, leading to weaning diarrhea. During this period, babies come in contact with the unhygienic environment and put their contaminated fingers in the mouth, making them susceptible to infection [63]. However, in a higher age group, children start adapting to food habits and develop better immunity, minimizing the risk of infection.

The detection of EAEC is a big challenge. HEp-2 cell assay, a gold standard procedure, is usually performed in analytical laboratories with cell culture facilities, needs expertise, and is time-consuming. Currently, there is a lack of consensus in the literature regarding which EAEC genes should be screened with PCR detection [64]. The identification of EAEC in this study was based on PCR with primers complementary for pCVD432, a test representing one of the most reliable means for detecting EAEC [65]. In studies conducted in Iran and Brazil, the same PCR assay performed with similar sensitivity (15–89%) and specificity (99%) [66, 67]. In a Swiss study investigating the pathogenic role of EAEC HIV-infected persons, the pCVD432 PCR assay was demonstrated to better correlate with clinical findings than the cell adherence assay [68]. Other studies, however, have provided evidence that pCVD432-probe-negative EAEC strains may also be associated with diarrheal illness [68]. An extreme example is represented by an outbreak of EAEC in Japan, where all isolates were negative by the pCVD432 PCR assay [69]. Several other reports have identified different virulence genes that play an important role in the pathogenesis of EAEC infection [70]. Though multiple efforts have been made to improve the detection of pathogenic EAEC strains by PCR based combinations of virulence genes, there is no single pathogenomic virulence marker. Therefore, here in the present study, we studied the distribution of virulence genes in EAEC isolated from asymptomatic and symptomatic children and also compared their distribution among typical vs. atypical strains.

Typical and atypical EAEC were defined as strains with or without *agg*R genes [71]. AggR is an EAEC transcription regulator that controls the expression of several putative virulence factors, including aggregate adherence fimbriae (AAF), dispersin, dispersin translocator Aat, and Aai type VI secretion. Morin et al. confirmed at least 44 AggR-regulated genes using DNA microarray and real-time quantitative reverse transcription-PCR (qRT-PCR) [71]. In our study, the *agg*R gene was identified in 66% and 33% of cases and control, respectively, and was significantly associated with diarrhea (p = 0.0001). The aEAEC isolates were more widespread in control (66%) than the diarrhoeal group (33%, p = 0.01). Our analysis was comparable with the Japanese study, where tEAEC (74.5%) was higher than aEAEC (25.5%) isolates [72]. These findings suggest that *agg*R-negative EAEC may be less significant in diarrhea pathogenesis. tEAEC isolates are more virulent than aEAEC since they bear more virulence genes, which include chromosomal (*aai*C, *air*, *eil*A, and *pic*) and plasmid genes (*aap*, ORF3, ORF61, *cap*U, *sat*, *pet*, and *ast*A). However, the correlation of the *agg*R gene with diarrhea is not consistent [36]. Huang and Sarantuya *et al.*, have found significant differences in *agg*R alone or combinations with other virulence genes in cases compared with controls [73]. While other reports did not observe any correlation [16, 74]. The lack of *agg*R in the majority of EAEC positive samples showed that this gene might not be a useful marker for EAEC diagnosis. PCR detection of *agg*R may not prove to be an appropriate initial screening test for EAEC, but it is informative because it can identify tEAEC, which is postulated to have a more pathogenic role than EAEC lacking *agg*R genes under *agg*R regulon [75].

In the present study, we found interesting results where strains lacking *agg*R gene possessed genes which are under the control of *agg*R regulon; this can be a topic of interest for further research why this happens. Virulence genes most prevalent in *agg*R negative isolates were *ast*A, ORF61, *cap*U, *rmo*A, and *esp*Y. This may be due to the mosaic nature of the EAEC genome or the presence of mutated plasmid in wild type strains.

Other genes that were more prominent in the diarrheal group than in the control group included the *aap* gene (p = 0.0006), which facilitates EAEC dispersal on the intestinal mucosa. In our study, *aap* gene found in 61.6% and 48.4% of EAEC in cases and controls, which is in contrast to the study from Mali, where this gene was present in 65% and 78.7% of cases and controls respectively [16].

The *ast*A gene codes EAST1 (EAEC heat-stable enterotoxins), causes increased chloride secretion and is correlated with secretory diarrhea [76]. Notably, in our study, the *ast*A gene was the most prevalent toxin among tEAEC and aEAEC isolates in the diarrheal group and was significantly associated with tEAEC causing diarrhea (p = 0.0005), indicating the pathogenetic function of this toxin. EAST1 is not limited to EAEC isolates, and very few studies have shown that this gene is correlated with diarrhea [77]. Stephen et al. found that the EAST1 genotype is not limited to EAEC but also identified from asymptomatic children in EHEC, other infective STEC, ETEC, and EPEC, as well as gastrointestinal *E. coli*, isolates [78].

The phylogenetic analysis in our study revealed that EAEC phylogeny is diverse and dispersed in all the phylogroups, similar to previous studies [79]. From the diarrheal group, most of the tEAEC and aEAEC strains belong to phylogroup D showing their ability to cause extraintestinal infection, whereas tEAEC and aEAEC strains from the non-diarrheal group belong to phylogroup B1 showing their commensal nature. This may indicate that EAEC originates from multiple lineages, as observed in the Nigerian study [80].

Several reports have documented a troubling high degree of multidrug resistance in EAEC strains. In our study, 75.8% of EAEC strains were MDR, which is higher than 50% reported from England. Resistance to ciprofloxacin, which is one of the most common agents to treat diarrhea, was 67.3% in the diarrheal group, similar to data from south India [81]. In our study, tEAEC isolates showed higher antibiotic resistance (ampicillin, ciprofloxacin, levofloxacin, cotrimoxazole, and ceftriaxone) than aEAEC. To the best of our knowledge, there have been no studies on aEAEC antibiotic susceptibilities from India [72]. The higher antibiotic resistance in EAEC may be due to the widespread use of antibiotics [82].

In conclusion, this study showed high EAEC prevalence among children from different regions of North India. The average incidence of diarrheal EAEC-positive samples was higher than the non-diarrheal community. Discrimination from carrier strains is essential for individual case treatment and epidemiological monitoring. The frequency of 21 separate virulence genes among these isolates demonstrated genetic heterogeneity of EAEC. The tEAEC isolates are more virulent than aEAEC. In this study, *ast*A, *aap*, and *agg*R genes were most frequently found in the EAEC from the diarrheal population. Within tEAEC, this gene combination is present in more than 50% of strains and helps differentiate tEAEC from aEAEC. This indicates that these alleles are EAEC variants and would be ideal as focus genes for EAEC identification, as well as tEAEC and aEAEC. Thus, we suggest using these plasmid-encoding genes as a signature sequence for EAEC identification. More studies aimed at clarifying whether the expressed proteins produced by the observed virulence genes play a role in the outcome of EAEC infection and pathogenesis are required. Identification of phenotype clusters (isolated or mixed genes) associated with both ill and healthy children indicates that pathophysiology of this enteric infection requires complex and dynamic regulation of several virulence genes. Most of EAEC strains from the diarrheal group belong to B2 and D phylogroups, which are potentially

pathogenic in nature. The high antibiotic resistance found in EAEC is a serious cause of concern.

## Supporting information

**S1 Fig.** Distribution of virulence-related markers among tEAEC and aEAEC in control A) and diarrheal group B). tEAEC: typical enteroaggregative *E. coli*, aEAEC: atypical enteroaggregative *E. coli*.
(TIF)

**S1 Table. Region-wise severity score.**
(DOCX)

**S2 Table. Distribution of EAEC virulence related markers among in diarrheal and control group.**
(DOCX)

**S3 Table. Combination of virulence markers among diarrheal and control EAEC.**
(DOCX)

## Acknowledgments

We acknowledge the kind support of local health authorities for facilitating the sample access in their areas.

## Author Contributions

**Conceptualization:** Vinay Modgil, Neelam Taneja.

**Data curation:** Vinay Modgil, Jaspreet Mahindroo, Manmohit Kalia.

**Formal analysis:** Vinay Modgil, Manmohit Kalia, Neelam Taneja.

**Funding acquisition:** Manish Kakkar, Balvinder Mohan, Neelam Taneja.

**Investigation:** Vinay Modgil, Jaspreet Mahindroo, Chandradeo Narayan, Md Yousuf, Varun Shahi, Meenakshi Koundal, Neelam Taneja.

**Methodology:** Vinay Modgil, Jaspreet Mahindroo, Chandradeo Narayan, Md Yousuf.

**Project administration:** Balvinder Mohan, Neelam Taneja.

**Resources:** Pankaj Chaudhary, Ruby Jain, Kawaljeet Singh Sandha, Seema Tanwar, Pratima Gupta, Kamlesh Thakur, Digvijay Singh, Neha Gautam, Manish Kakkar, Bhavneet Bharti.

**Software:** Vinay Modgil.

**Supervision:** Bhavneet Bharti, Balvinder Mohan, Neelam Taneja.

**Validation:** Vinay Modgil, Neelam Taneja.

**Visualization:** Vinay Modgil, Neelam Taneja.

**Writing – original draft:** Vinay Modgil, Manmohit Kalia, Neelam Taneja.

**Writing – review & editing:** Vinay Modgil, Manmohit Kalia, Manish Kakkar, Neelam Taneja.

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
