## [Decision Letter · Decision Letter 0]

29 Jun 2020

Dear DR. Taneja,

Thank you very much for submitting your manuscript "Comparative analysis of virulence determinants, phylogroups and antibiotic susceptibility patterns of typical versus atypical Enteroaggregative E. coli in India" for consideration at PLOS Neglected Tropical Diseases. As with all papers reviewed by the journal, your manuscript was reviewed by members of the editorial board and by several independent reviewers. In light of the reviews (below this email), we would like to invite the resubmission of a significantly-revised version that takes into account the reviewers' comments. 

We cannot make any decision about publication until we have seen the revised manuscript and your response to the reviewers' comments. Your revised manuscript is also likely to be sent to reviewers for further evaluation.

Sincerely,

Husain Poonawala

Guest Editor

Ana LTO Nascimento

Deputy Editor

Reviewer's Responses to Questions

**Key Review Criteria Required for Acceptance?**

**Methods**

-Are the objectives of the study clearly articulated with a clear testable hypothesis stated?

-Is the study design appropriate to address the stated objectives?

-Is the population clearly described and appropriate for the hypothesis being tested?

-Is the sample size sufficient to ensure adequate power to address the hypothesis being tested?

-Were correct statistical analysis used to support conclusions?

-Are there concerns about ethical or regulatory requirements being met?

Reviewer #1: Lines 176-182: More detailed methods need to be provided for the selective culturing and identification of the E. coli and other diarrheal pathogens described.

Reviewer #2: Methods

-Are the objectives of the study clearly articulated with a clear testable hypothesis stated?: 

Yes

-Is the study design appropriate to address the stated objectives?

Yes

-Is the population clearly described and appropriate for the hypothesis being tested?

Yes

-Is the sample size sufficient to ensure adequate power to address the hypothesis being tested?

Yes

-Were correct statistical analysis used to support conclusions?

Yes

-Are there concerns about ethical or regulatory requirements being met?

No

Reviewer #3: The objectives are clearly stated and the study design is appropriate. 

For the population studied in the control group: did the authors follow up if the children that tested positive showed or had previously shown any symptoms? Does infection in control group sample mean presence of bacteria or presence of symptoms. How long after or before the symptoms set in can the bacteria be detected in the stool sample? 

Are the antibiotics tested here the first line of treatment given to children showing symptoms?

**Results**

-Does the analysis presented match the analysis plan?

-Are the results clearly and completely presented?

-Are the figures (Tables, Images) of sufficient quality for clarity?

Reviewer #1: Fig 1: How are the sampling sites indicated? Please describe.

Table 1: Are these the primers that were used for pathotype classification and are described elsewhere in the manuscript as M-PCR? Please clarify.

Fig 3: Please provide greater detail in the figure legend, describing what is in each lane. The blue arrows stretching across the image are difficult to see, and some of the words in red next to the gel are smaller and difficult to read.

Table 9: These are both tEAEC and aEAEC?

Tables 9, 10, and 11: These tables could be combined to save space in the manuscript.

Table 12: This table could be moved to supplemental data.

Figures 6 and 7: These figures could be combined to make it easier to compare virulence gene trends between diarrheal vs. control EAEC. 

Tables 13, 14, and 15: These tables should be combined for ease of comparison.

Figure 8: This figure could be removed as it is redundant with the tables.

Tables 16, 17, and 18: These tables should be combined for ease of comparison. Also, it’s not clear why Table 16 has colors.

Reviewer #2: -Does the analysis presented match the analysis plan?

Yes

-Are the results clearly and completely presented?

Please see below

-Are the figures (Tables, Images) of sufficient quality for clarity?

The study is very interesting and has been elaborately performed. However, the authors should try to make the presentation more concise.

 1) A number of figures are duplicating the information given in the Tables. This should be avoided

 2) Fig 2 shows information on 273 DEC positive children. Hence, the bar showing DEC strains may be omitted since the total number of DEC can be made out from the other bars which depict subsets of DEC.

 3) Table 3- The percentage of EAEC etc has been calculated out of 273 DEC, not the total number of patients i.e.1210. However, in the break up of ages of patients, the total has been calculated out of total number of patients. A uniform method should be used in the table.

 4) Fig 4 is not clear as the legend says "Distribution of DEC groups by age in children without diarrhea". However, the ages cannot be made out in the figure. Since the same information is seen in Table 4, Fig 4 can be omitted.

 5) Table 5 gives "Distribution of DEC pathotypes among diarrheal and healthy children". Hence the total number of patients need not be mentioned in the Table. This confuses the reader.

 6)Fig 5 is not needed. The information given in the text is sufficient.

 7) Fig 6 is not needed. Table 10 gives the same information. Table 10 and 11 can be combined for better comparison.

 8) Similarly Fig 7 is repetitive of Table 11.

 9) Fig 8 is repetitive.

 10) Fig 14 and 15 can be combined.

Reviewer #3: Table 3: The data is distributed 0-2, 2-5 and <5. Have the authors analysed if 0-2 and 2-5 are statistically different, before pooling in the data for analysis. 

Table 4: The groups for age distribution should be kept the same for comparable analysis. 

Table 5: The authors need to show data only from DEC positive samples and not overall samples. Please carry out statistical analysis for EAEC positive samples in control vs diseases group. 

Table 8: is tEAEC and aEAEC significant in either diarrheal or the control group?

**Conclusions**

-Are the conclusions supported by the data presented?

-Are the limitations of analysis clearly described?

-Do the authors discuss how these data can be helpful to advance our understanding of the topic under study?

-Is public health relevance addressed?

Reviewer #1: (No Response)

Reviewer #2: -Are the conclusions supported by the data presented?

Yes

-Are the limitations of analysis clearly described?

No

-Do the authors discuss how these data can be helpful to advance our understanding of the topic under study?

This can be elaborated

-Is public health relevance addressed?

Yes

Reviewer #3: (No Response)

**Editorial and Data Presentation Modifications?**

Reviewer #1: (No Response)

Reviewer #2: Minor revision

Reviewer #3: All the figures and legends require more description. 

Fig2,4: Do not add DEC data as a bar in the graph as the authors are showing distribution of subDEC strains. 

Table 6: can be removed or moved to supplements as it discusses severity of non-DEC positive infections. 

Table 9, 10, 11, 12 and its associated figures need to be condensed and represented differently, focusing on major findings. 

Table 13, 14, 15 and fig8: also needs to be represented differently in a condensed format

Table 16, 17, 18 also needs to be condensed and needs removal of redundant data.

**Summary and General Comments**

Reviewer #1: The manuscript titled “Comparative analysis of virulence determinants, phylogroups and antibiotic susceptibility patterns of typical versus atypical Enteroaggregative E. coli in India” describes the prevalence of EAEC compared to other diarrheagenic E. coli pathotypes among diarrheal versus non-diarrheal stools of children in North India. The authors further described the EAEC isolates by classifying them as tEAEC or aEAEC and comparing the virulence gene content of these sub-groups of EAEC among the diarrheal versus non-diarrheal stools.

Throughout the manuscript there are missing spaces between words.

Lines 71-72: Please describe what the CVD432 probe is.

Line 75: Please write out PGI.

Lines 77-78: Was gene expression examined? Or were the genes detected using PCR?

Lines 79-81: It is not clear what is meant by “our signature sequence”. Are you referring to the combination of the three genes astA, aap, and aggR. Are you saying the detection of these genes can be used to identify both tEAEC and aEAEC? This is not clear.

Line 81: Write out MDR, and clarify what is meant by “more resistant”. Do the tEAEC typically exhibit resistance to a greater number of antibiotics? Is there an average number of antibiotics that the tEAEC exhibited reduced susceptibility to compared to the aEAEC?

Lines 108-109: Please clarify the sentence “Clinical features of EAEC diarrhea are usually well-defined in sporadic cases outbreaks and voluntary model.”. It is not clear what is meant by sporadic cases. Or what is meant by voluntary model. Also, please describe the clinical features.

Lines 133-136: The revised classification scheme by Jaureguy et al. (PMID: 19036134) described 5 lineages of E. coli (A+B1, B2, D, E, and F). The statement that most DEC strain belong to phylogroup D, while commensals belong to phylogroups A and B1 is not correct. ETEC in particular is predominantly identified in phylogroups A and B1, while EPEC is most often identified in phylogroups B2 and B1.

Lines 163-164: Please provide a reference for the Vesikari score. 

Line 189: Please describe the region amplified by the CVD432 primers.

Lines 204-206: Were the aEAEC identified only by the absence of aggR? Or they also contained the CVD432 region? Were they verified phenotypically to be EAEC by their adherence patterns in HEp-2 cell culture?

Line 242: This includes both tEAEC and aEAEC?

Lines 572-573: The sentence starting with “Many EAEC strains….” is not clear.

Reviewer #2: This is a really extensive piece of work and must be shared with the scientific community. However, the data presentation needs to fine tuned. There is a lot of repetition. The paper should be made concise for better understanding

Reviewer #3: (No Response)

PLOS authors have the option to publish the peer review history of their article (what does this mean?). If published, this will include your full peer review and any attached files.

Reviewer #1: No

Reviewer #2: No

Reviewer #3: No
---

## [Decision Letter · Decision Letter 1]

1 Sep 2020

Dear Dr. Taneja,

We are pleased to inform you that your manuscript 'Comparative analysis of virulence determinants, phylogroups and antibiotic susceptibility patterns of typical versus atypical Enteroaggregative E. coli in India' has been provisionally accepted for publication in PLOS Neglected Tropical Diseases.

Best regards,

Husain Poonawala

Guest Editor

Ana LTO Nascimento

Deputy Editor

Reviewer's Responses to Questions

**Key Review Criteria Required for Acceptance?**

**Methods**

-Are the objectives of the study clearly articulated with a clear testable hypothesis stated?

-Is the study design appropriate to address the stated objectives?

-Is the population clearly described and appropriate for the hypothesis being tested?

-Is the sample size sufficient to ensure adequate power to address the hypothesis being tested?

-Were correct statistical analysis used to support conclusions?

-Are there concerns about ethical or regulatory requirements being met?

Reviewer #1: (No Response)

Reviewer #2: The authors have modified the Methods section satisfactorily.

Reviewer #3: Objectives are clearly articulated with clear testable hypothesis

**Results**

-Does the analysis presented match the analysis plan?

-Are the results clearly and completely presented?

-Are the figures (Tables, Images) of sufficient quality for clarity?

Reviewer #1: (No Response)

Reviewer #2: The authors have modified the Results section satisfactorily.

Reviewer #3: The results and analysis in the revised manuscript are clearly presented. the condensed version of the tables give clarity to the manuscript

**Conclusions**

-Are the conclusions supported by the data presented?

-Are the limitations of analysis clearly described?

-Do the authors discuss how these data can be helpful to advance our understanding of the topic under study?

-Is public health relevance addressed?

Reviewer #1: (No Response)

Reviewer #2: Data is supported

Reviewer #3: Authors have made relevant conclusions

**Editorial and Data Presentation Modifications?**

Reviewer #1: (No Response)

Reviewer #2: A shorter manuscript would improve its readability.

Reviewer #3: (No Response)

**Summary and General Comments**

Reviewer #1: (No Response)

Reviewer #2: The authors have modified the manuscript satisfactorily. The discusison can be shortened and made more crisp.

Reviewer #3: Overall, the revised version and the answers to the reviewers comments are satisfactory

PLOS authors have the option to publish the peer review history of their article (what does this mean?). If published, this will include your full peer review and any attached files.

Reviewer #1: No

Reviewer #2: No

Reviewer #3: No

---

## [Editor Report · Acceptance letter]

20 Oct 2020

Dear Dr Taneja,

We are delighted to inform you that your manuscript, "Comparative analysis of virulence determinants, phylogroups and antibiotic susceptibility patterns of typical versus atypical Enteroaggregative E. coli in India," has been formally accepted for publication in PLOS Neglected Tropical Diseases.

Best regards,

Shaden Kamhawi

co-Editor-in-Chief

Paul Brindley

co-Editor-in-Chief
